# Effects of the COVID-19 Pandemic on Job Activity, Dietary Behaviours and Physical Activity Habits of University Population of Naples, Federico II-Italy

**DOI:** 10.3390/ijerph18041502

**Published:** 2021-02-05

**Authors:** Mariarita Brancaccio, Cristina Mennitti, Alessandro Gentile, Luca Correale, Cosme Franklim Buzzachera, Cinzia Ferraris, Cristina Montomoli, Giulia Frisso, Paola Borrelli, Olga Scudiero

**Affiliations:** 1Department of Molecular Medicine and Medical Biotechnology, University of Naples Federico II, Via S. Pansini 5, 80121 Naples, Italy; brancacciomariarita2@gmail.com (M.B.); cristinamennitti@libero.it (C.M.); alexgenti98@libero.it (A.G.); 2Department of Public Health, Experimental and Forensic Medicine, Unit of Motor Science, University of Pavia, Via Forlanini 2, 27100 Pavia, Italy; luca.correale@unipv.it (L.C.); cosme.buzzachera@unipv.it (C.F.B.); 3Department of Public Health, Experimental and Forensic Medicine, Laboratory of Food Education and Sport Nutrition, University of Pavia, Via Agostino Bassi 21, 27100 Pavia, Italy; cinzia.ferraris@unipv.it; 4Department of Public Health, Experimental and Forensic Medicine, Unit of Biostatistics and Clinical Epidemiology, University of Pavia, Via Forlanini 2, 27100 Pavia, Italy; cristina.montomoli@unipv.it; 5Ceinge Biotecnologie Avanzate S. C. a R. L., 80131 Naples, Italy; 6Department of Medical, Oral and Biotechnological Sciences, Laboratory of Biostatistics, University G. d’Annunzio of Chieti-Pescara, Via dei Vestini 31, 66100 Chieti, Italy; 7Task Force on Microbiome Studies, University of Naples Federico II, 80100 Naples, Italy

**Keywords:** COVID-19, eating habits, physical activity, health, survey and university population

## Abstract

Coronaviruses (CoVs) are a large family of respiratory viruses that can cause mild to moderate illness. The new variant COVID-19 has started to spread rapidly since December 2019, posing a new threat to global health. To counter the spread of the virus, the Italian government forced the population to close all activities starting from 9 March 2020 to 4 May 2020. In this scenario, we conducted a cross-sectional study on a heterogeneous sample (average age of 28 ± 12 years, 62.6% females) of the University of Naples Federico II (Italy). The aim of the study was to describe the lifestyle change in the university population during quarantine for the COVID 19 pandemic. Participants compiled an online survey consisting of 3 sections: socio-demographic data, dietary behaviours, physical activity habits and psychological aspects. The different results by gender are: 90.8% of females continued to work from home (81.9% were students); 34.8% increased their physical activity; and, only 0.8% prefer ready meals. Whereas, the same percentage of men continued to work from home (90%), but only 72.1% were students (*p* < 0.001 vs. females), only 23.9% increased physical activity (*p* < 0.001) and 1.7% favous ready meals. Our data shows that the male population was more affected by isolation and quarantine reporting more unfavourable behavioural changes.

## 1. Introduction

On 11 March 2020 the World Health Organization has declared “global epidemic” due to the spread of the disease from coronavirus 2019 (COVID-19) [1]. Quickly, the virus started extent to Asia, Europe, North America and the Middle East. New cases and outbreaks of COVID-19 have also been reported in Africa and the South America. According to data from the Center for Systems Science and Engineering of the Johns Hopkins Whiting School of Engineering, on 25 December 2020, the number of cases has reached 80.4 million across the world, and the death toll has risen to over 1.76 million people. The United States currently has the highest number of cases worldwide (19 million), with over 332,000 victims. Unlike, in Europe, according to the world health organization (WHO), Italy is in fourth place in terms of the number of cases (2.4 million), but in the first place as a mortality rate (about 3%) [2]. The reasons behind the rapid spread of COVID-19 in Italy have not yet been fully understood [3].

To avoid the collapse of the Italian health system, on 9 March 2020, the Italian government declared quarantine for the entire country and instituted self-isolation between the regions [4]. The combination of containment measures and the slowdown in the spread of the virus made it possible to treat as many patients as needed intensive care and at the same time to protect the most vulnerable sections of the population.

Containment measures, including self-isolation and physical distancing, were still in place more stringent in some regions of Italy. On 22 March 2020, the Campania government and the local health authorities have imposed further restrictions on public movement and banned sports activities and exercise outdoors. Residents of the Campania region were allowed to train outdoors nearby their habitations and to walk the dog without departing more than 200 m from home. These restrictions, essential to contain the spread of the COVID-19 disease [5], may, in some circumstances, have limited the adoption or maintenance of behaviour health benefits whit unfavourable repercussions on citizens. 

In fact, the Coronavirus emergency has led to an increase in sedentary life in both childhood and adulthood [6,7], underlining how the pandemic has influenced the lifestyle of the world population [8,9,10]. Certainly, the social distancing and the restrictions applied during the quarantine period have upset the habits of workers and students. In this regard, the National Bureau of Economic Research reported that between February and May 2020 30% of American workers switched to smart working (working from home WFH) and that 10% were fired. This drastic change cause an increase in sedentary lifestyle [11]. To assess the impact of COVID-19 in the student population, a group of researchers led by *Aucejo* conducted an analysis of 1500 students from the University of Arizona. This investigation revealed that 13% of students have delayed graduation, 40% have lost a job, internship, or job offer, and 29% expect to earn less at age 35 [12]. In addition, in the United States, all clinical learning has been suspended indefinitely in accordance with the recommendations of the American Association of Medical Colleges (AAMC) of 17 March 2020. This change forced medical students to move away from hospitals and follow virtual classes. A study by *Byrnes* et al. highlighted the students’ concerns regarding their preparation and future career [13]. 

Home confinement has also led to a change in eating habits. Some studies have revealed that people have increased the consumption of unhealthy foods, eating out of control, more snacks between meals and an overall greater number of main meals. [14,15,16]. On the other hand, a study conducted in Spain highlighted the beneficial effects that the Mediterranean diet had on the population. In fact, the increase in the consumption of fruit, vegetables or legumes and a lower intake of red meat, alcohol, fried or sweet foods than usual, could have a positive impact on the prevention of chronic diseases and complications related to COVID-19 [17].

The quarantine period also had different effects on sports practice in the various countries [18,19,20]. In one Bavarian population, an increase in physical activity was reported during the COVID-19 pandemic, as a large-scale blockade was not imposed in Bavaria like those in France, Spain and Italy. Conversely, Germany allowed outdoor exercise, but only alone or with other family members [21].

This study aimed to evaluate the effects of quarantine on the eating and physical activity habits of a heterogeneous university population of the Campania region and to detect gender differences. Through statistical analysis, we underline how modern society has a deficit that should be filled to ensure a healthy lifestyle even during a global emergency.

## 2. Materials and Methods

### 2.1. Formulation and Administration of the Questionnaire

A cross-sectional study has been carried out. The questionnaire, built by a group of researchers and academics from the University of Pavia experts in public health, sports science, dietary behaviour and psychology, was administered anonymously through an online platform to the university population (students, teaching and technical and administrative staff) of the University of Naples Federico II in May 2020.

Information on the study, objectives and participation links were communicated by e-mail from the Administration to the university population and participants (mean age 28.4 ± 11.9 years, range 18–75) after viewing agreed to complete the questionnaire. Participants were asked to answer the questions of the self-administered online questionnaire (filling in time of less than 15 min) informing them that they could interrupt the compilation at any time, without the obligation to justify the decision. The study was conducted in accordance with the Declaration of Helsinki and the data have been processed in compliance with the current privacy law (EU Regulation 2016/679 and Privacy Code D.Lgs. 101/2018). Because we carried out an online anonymized survey no formal agreement to the ethical committee has been requested.

The “Lifestyle and Psychological Well-being Questionnaire during Emergency by COVID-19” consists in 48 items, with single/multiple choice answers, to assess lifestyle changes in the university population induced by the quarantine period. The questionnaire was not validated.

Specifically, the first section (21 items) assesses socio-demographic data (age, gender, weight before of quarantine and current, height, smoking habits, number of cigarettes per day, marital status, education and position in the University) health and diseases status (test positive for COVID-19 infection, presence of flu symptoms and fever over 38.5 °C and number of days), type of dwelling (type of apartment, house with garden and number of roommates) and job activity before and during quarantine.

Dietary behaviours and physical activity habits were described in the second section (22 items) and included question on drink alcoholic beverages frequency, number of meals consumed before and during quarantine, consumption of more food than before quarantine, consumption of fruit and vegetables during quarantine, increased consumption of convenience food and fast food compared to the period before quarantine, improvement of diet and food quality compared to the period before quarantine, have breakfast, food consumption during the day without hunger, frequency of aperitif consumption and amount of water drunk per day. While for the physical activities were investigated sports activities carried out before quarantine, time spent doing moderate or vigorous activities during the week, frequency of exercise before quarantine, previous level of physical activity compared to period of quarantine, current exercise frequency during the quarantine, objective to be achieved through physical activities, exercises using tutorial on internet, train alone during the quarantine and influence of quarantine on time spent sitting or lying down.

Finally, the third section (6 items) contains data on psychological aspects relating to the measurement of distress resulting from depressive and anxious symptoms in the last 30 days. The following results are based on the description of the first and second sections of the questionnaire.

### 2.2. Statistical Analysis

Descriptive analysis was carried out using means and standard deviation or median and interquartile range (IQR) for the quantitative variables and percentages values for the qualitative ones. Normality distribution for quantitative variables was assessed by the Shapiro-Wilk. Univariate comparisons were investigated between groups and explicative variables using the Pearson χ2 test or the Fisher’s exact test for categorical data, and the Student’s *t*-test for independent data or non parametric Wilcoxon rank-sum test when appropriate for continuous data. Statistical significance was taken at the <0.05 level. All analyses were performed using STATA software v15.1 (StataCorp, College Station, Texas, USA).

## 3. Results

### 3.1. Characterization of the Group under Examination

A total of 1130 individuals attending the University Federico II completed the survey in May 2020. 62.6% (*n* = 707) of these were females with mean age 27.4 ± 11 (females) and 30.1 ± 13 years (males). In addition, 78.2% of respondents were students, 11.0% administrative staff and 10.8% were faculty members.

Table 1 shows the characteristics of the population and its stratification by gender. We found statistically significant differences based on gender regarding the following variables: age, weight before and during quarantine, height, BMI before quarantine and current.

In particular, during the quarantine both females and males underwent weight gain (see Table 1); in two months the mean weight gain was 0.5 and 0.7 kg respectively in females and males; as a consequence, both females and males reported an increase in BMI equal to 0.2 while maintaining a normal weight condition. Furthermore, males consumed more cigarettes than females (see Table 1). In addition, again according to a gender stratification, from Table 1 it emerges that males became ill with COVID-19 more than females (0.5% and 0.1%, respectively).

### 3.2. Type of Dwelling

No statistically significant differences were found for apartment type, having/not having a garden or number of people living together (Table 2). 

### 3.3. The Consequences of Quarantine on Job Activity

To examine the effects of the pandemic on work activities, we stratified the population according to the qualification and the position held in the university, that is professors, administrative staff and students We found statistically significant differences between male and female regarding education (χ^2^ = 10.53, degree of freedom *df* = 3, *p* = 0.012) and position in the University (χ^2^ = 24.26, *df* = 3, *p* < 0.001) see Figure 1 However, there was no statistically significant difference based on gender regarding the work/study habit adopted during the quarantine (Table 3).

### 3.4. The Pandemic’s Effect on Dietary Behaviours

To highlight if and what effects the quarantine had on the dietary behaviours of the population in question, we asked these individuals for information about their type of diet.

First, in Figure 2A,B we have reported some eating habits of the stratify population examined. In particular, in Figure 2A we highlighted if the quantity of food consumed had increased; in Figure 2B what was the origin of the food consumed and also, we analysed the differences between males and females regarding the origin of cooked and/or eaten food.

In addition, in Table 4 showed the parameters considered and statistically significant differences based on gender regarding drinking alcoholic beverages, amount of meals during the period in question, consuming more food than before the quarantine, improved diet and food quality compared to the period before quarantine, have breakfast and amount of water drunk per day.

### 3.5. Physical Activity before and during the Lockdown

To investigate the effects of the lockdown on physical activity habits, we asked participants to answer questions about the type and intensity of exercise and daily physical activity they were used to do before the quarantine period. Then, we asked if these habits had changed during the lockdown. 

First of all, we considered the changes in physical activity on the total population examined, in this case we highlighted an increase of behaviours sedentary as shown in Figure 3.

We then examined the data obtained through gender stratification (see Table 5). In this case the results showed that differences in gender were associated with all the physical activity habits except for time spent doing vigorous activities during the week, train alone during quarantine and the influence of the quarantine period on sedentariness (Table 5). 

## 4. Discussion

COVID-19 is responsible for a global pandemic with millions of deaths and infected people [22]. In Italy, to reduce the spread of the virus and the collapse of the health system, the government applied restrictive measures that became increasingly stringent until 9th March 2020, the Italian government declared the quarantine for the entire Nation. On one hand, for sure these measures allow to fight the spread of the virus, on the other, it is certainly accompanied by negative consequences.

Quarantine is often an unpleasant experience for those who are living: separation from loved ones, loss of freedom, job uncertainty, loneliness and fear of illness can all have dramatic effects [23].

Social distancing, the closure of the schools and numerous work activities, the ban on group meetings and restrictions on physical activities have abruptly upset the traditional lifestyle [24,25].

Our study for the first time analysed the fundamental aspects of every citizen’s daily life such as work, physical activity and eating habits.

From the data in our hands, males have a greater weight gain and a higher consumption of cigarettes than females; factors that could partly explain their greater predisposition to contract COVID-19. 

The fact that men are more likely than females to get sick is a fact also highlighted in other studies conducted following the appearance of COVID-19. Most likely, the incidence of the disease among males is to be attributed to various factors such as the immune system, hormonals modulation, cardiovascular diseases, eating habits, alcohol consumption and smoking [26,27].

Alongside these results, the analysis of the habits of life of the university population highlighted that males compared to females during the quarantine significantly intensified the consumption of take-away food and alcohol. These data support weight gain which is the manifestation of a bad and unbalanced diet and also, support the increased frequency of COVID-19 occurrence among the male population [26,27].

These results are similar to other studies conducted in the COVID-19 period in other countries. For instance, the quality of Canadian students’ diet was poorer during COVID-19 given the decreased frequency of consumption of grains, fruits, vegetables, dairy, nuts, meat and meat alternative while alcohol consumption increased significantly [28]. Furthermore, a survey conducted by thirty-five research organizations from Europe, North-Africa, Western Asia and the Americas reported that food consumption and meal patterns (the type of food, eating out of control, snacks between meals, number of main meals) were more unhealthy during confinement, with only alcohol binge drinking decreasing significantly [14]. Contrariwise, COVID-19 confinement in Spain has led to the adoption of healthier dietary behaviors in the studied population, as reflected by a higher adherence to the Mediterranean Diet [16]. Moreover, the COVID-19 pandemic may have changed the food choice determinants of Polish adolescents, as it may have increased the importance of health and weight control, but reduced the role of mood and sensory appeal. This may be interpreted as positive changes promoting the uptake of a better diet than in the period before the pandemic [12].

Finally, we shed light on exercise and physical activity habits. First of all, by analysing the entire population, we highlighted an increase in sedentary lifestyle (Figure 3); most likely, this increase is caused by the concomitance of performing sedentary jobs and spending many more hours ahead of the PC (smart-working). Similar data was reported by *McDowell* and co-workers. In particular, they studied an American workers population (2,303 adults) and pointed out that about 95% had a more sedentary life than before, and that this sedentary lifestyle was also related to the type of work [11].

Then, through gender stratification, we pointed out that females practiced more consistent workouts during the quarantine period (see Table 5) while taking care of their bodies. This observation probably highlights that inadequate physical activity during the pandemic caused weight gain in males.

In this scenario, through the statistical analysis conducted, we can affirm that males have become brutalized compared to females, suffering significantly from quarantine and the deprivations that are caused by it, manifesting their discomfort in weight gain, accompanied by high intake of food, to a reduction in physical activity which then culminated in an increase in alcohol consumption. 

All these factors acted as substrate for a weakening of the immune system mostly of man which made them more predisposed to contract COVID-19 [27].

In addition, obesity and sedentariness represent two fundamental factors in the onset of pathologies, such as cardiovascular [29,30] and respiratory disease [31]. During the quarantine, it has been shown that obese patients are exposed to an increased risk of severe respiratory, even if the link between the obesity and the severity of COVID-19 is unclear [32]. The complete absence of physical activity and an incorrect diet can cause weight gain and related pathologies. In fact, it has been shown that the lockdown has also led to an increase in diseases such as diabetes and hypertension [33,34].

The beneficial effects of exercise are well known [35]. It has been widely demonstrated that physical exercise can prevent infections [36,37,38], inflammation and delay the onset of cardiovascular and muscular pathologies [39,40]. In this scenario, the practice of physical activity conducted according to the ANTI-COVID-19 rules, and a correct diet, such as Mediterranean Diet can represent a valid tool to combat weight gain and depressive moods that can characterize periods such as those of quarantine.

This study has several limitations. The cross-sectional design not provide any information on the possible causal nature and the associations identified may be difficult to interpret. In addition, self-reported data are susceptible to biases information. Finally, the use of non-validated questionnaire raises the problem of generalization of results.

## 5. Conclusions

The administration of this questionnaire within the Campanian university population showed how the restrictive measures applied during the COVID-19 pandemic have influenced the usual lifestyle. Changes in the lifestyle of people including physical activity and eating habits can have repercussions on physical and psychological health, as well as affect the well-being of citizens in the future. Therefore, these statistical studies must be the springboard for functional studies that will be needed when such containment measures will be completely eliminated.

## Figures and Tables

**Figure 1 ijerph-18-01502-f001:**
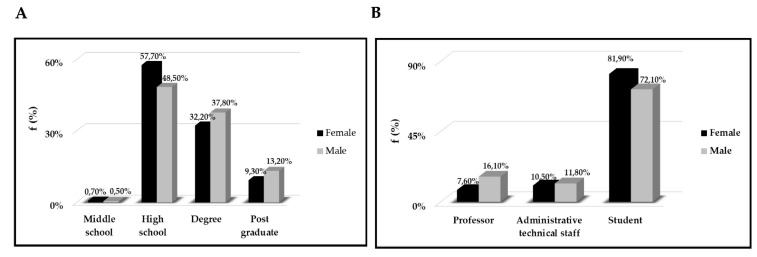
(**A**) Representation of the educational level of the population examined. On the y-axis we reported the frequency as a percentage (%) respect to stratification by gender, and on the x-axis. The degree of education. (**B**) Stratification according to gender of the role covered within the university of the population examined. On the y-axis we reported the frequency as a percentage (%) respect to stratification by gender, and on the x-axis. The job activity.

**Figure 2 ijerph-18-01502-f002:**
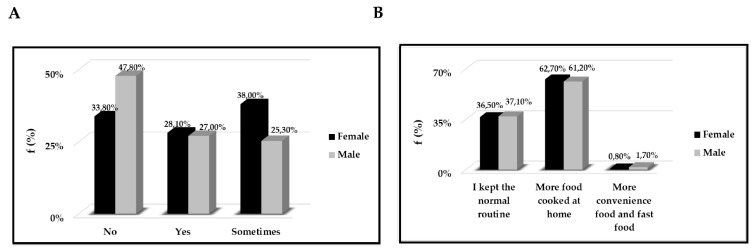
(**A**) Dietary behaviours. On the y-axis we reported the frequency as a percentage (%) respect to stratification by gender, and on the x-axis if the amount of food consumed during the quarantine had increased, remained unchanged or decreased (**B**) Quality of food used. On the y-axis we reported the frequency as a percentage (%) respect to stratification by gender, and on the x-axis the types of food used during the quarantine.

**Figure 3 ijerph-18-01502-f003:**
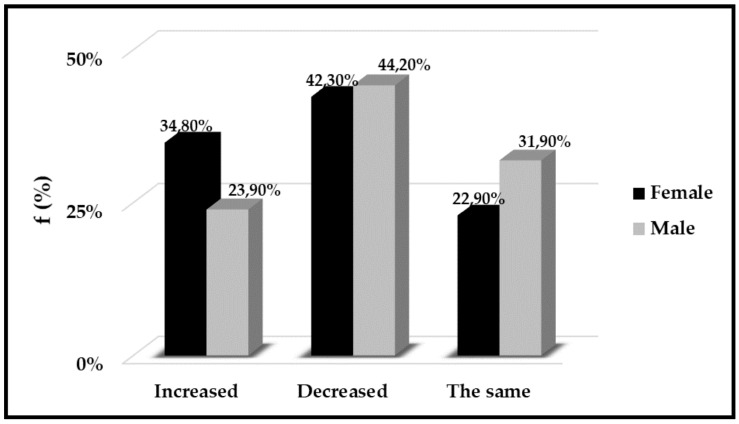
Physical activity during quarantine. On the y-axis we reported the frequency as a percentage (%) to the total population and on the x axis the variation of physical activity habits.

**Table 1 ijerph-18-01502-t001:** Characteristics of the university population and stratification by gender.

	Total *N* = 1130	Female *N* = 707	Male *N* = 423	*p*-Value
**Age (years)**	28.4 ± 11.9	27.4 ± 11.0	30.1± 13.0	<0.001 *
**Weight (kg) before the beginning of quarantine**	67.4 ± 14.9	61.4 ± 12.2	77.4 ± 13.5	<0.001 *
**Current weight (kg)**	68.0 ± 15.2	61.9 ± 12.5	78.1 ± 13.9	<0.001 *
**Height (cm)**	169.1 ± 9.5	164.3 ± 6.1	177.2 ± 8.8	<0.001 *
**BMI (kg/m^2^)** **before quarantine**	23.4 ± 4.3	22.7 ± 4.3	24.6 ± 4.1	<0.001 *
**BMI (kg/m^2^) current**	23.6 ± 4.4	22.9 ± 4.3	24.8 ± 4.3	<0.001 *
**Do you smoke?**				
*No*	929 (82.2%)	592 (83.7%)	337 (79.7%)	0.084 ***
*Yes*	201 (17.8%)	115 (16.3%)	86 (20.3%)	
**If yes: how many cigarettes per day?**	5 (2–10)	5(2–9)	7(2–12.5)	0.170 **
**Marital Status**				
*Married/cohabitant*	361 (31.9%)	221 (31.3%)	140 (33.1%)	0.520 ***
*Single*	769 (68.1%)	486 (68.7%)	283 (66.9%)	
**Did you test positive for COVID-19 infection?**				
*No*	168 (14.9%)	102 (14.4%)	66 (15.6%)	0.490 ****
*I did not take the test*	959 (84.9%)	604 (85.4%)	355 (83.9%)	
*Yes*	3 (0.3%)	1 (0.1%)	2 (0.5%)	
**Have you had flu symptoms? (cold, fever, cough ecc)**				
*No*	1013 (89.6%)	635 (89.8%)	378 (89.4%)	0.810 ***
*Yes*	117 (10.4%)	72 (10.2%)	45 (10.6%)	
**If yes, have you had fever over 38.5 °C?**				
*No*	620 (99.0%)	383 (99.5%)	237 (98.3%)	0.150 ****
*Yes*	6 (1.0%)	2 (0.5%)	4 (1.7%)	
**If yes, how many days?**	2 (1–3)	2.5 (0.5–3)	2 (1–3.5)	0.896 **

*N* (%) or mean and standard deviation or median (IQR) are shown when appropriate; * Student *T* test for independent data or ** Wilcoxon rank-sum test; *** Pearson’s chi-square test or **** Fisher’s exact test.

**Table 2 ijerph-18-01502-t002:** Characteristics of dwelling of the university population and stratification by gender.

	Total *N* = 1130	Female *N* = 707	Male *N* = 423	*p*-Value
**Indicate the type of your apartment**				
*Two-room apartment*	95 (8.4%)	62 (8.8%)	33 (7.8%)	0.820 ***
*One-room apartment*	31 (2.7%)	20 (2.8%)	11 (2.6%)	
*Three-room apartment or more*	1004 (88.8%)	625 (88.4%)	379 (89.6%)	
**Does your house have a garden?**				
*No*	561 (49.6%)	339 (47.9%)	222 (52.5%)	0.140 ***
*Yes*	569 (50.4%)	368 (52.1%)	201 (47.5%)	
**How many people live in your house apart from you?**	3 (2–3)	3 (2–3)	3 (2–3)	0.185 **

*N* (%) or mean and standard deviation or median (IQR) are shown when appropriate; ** Wilcoxon rank-sum test; *** Pearson’s chi-square test or **** Fisher’s exact test.

**Table 3 ijerph-18-01502-t003:** The quarantine’s effect on job activity of the university population and stratification by gender.

	Total *N* = 1130	Female *N* = 707	Male *N* = 423	*p*-Value
**Before the quarantine, was your work/study in Hospital setting?**				
*No*	821 (72.7%)	511 (72.3%)	310 (73.3%)	0.710 ***
*Yes*	309 (27.3%)	196 (27.7%)	113 (26.7%)	
**During the quarantine:**				
*I continued studying/working from home*	1040 (92.0%)	656 (92.8%)	384 (90.8%)	0.480 ***
*I continued studying/working normally leaving home*	56 (5%)	32 (4.5%)	24 (5.7)	
*I stopped studying/working*	34 (3%)	19 (2.7%)	15 (3.5%)	

*N* (%) or mean and standard deviation are shown when appropriate; *** Pearson’s chi-square test.

**Table 4 ijerph-18-01502-t004:** Eating habits before and during the pandemic period of the university population and stratification by gender.

	Total *N* = 1130	Female *N* = 707	Male *N* = 423	*p*-Value
**Do you drink alcoholic beverages?**				
*Daily*	29 (2.6%)	10 (1.4%)	19 (4.5%)	<0.001 ***
*1–2 times a week*	152 (13.5%)	86 (12.2%)	66 (15.6%)	
*3–4 times a week*	40 (3.5%)	20 (2.8%)	20 (4.7%)	
*Occasionally*	647 (57.3%)	406 (57.4%)	241 (57.0%)	
*Never*	262 (23.2%)	185 (26.2%)	77 (18.2%)	
**How many meals did you eat before the quarantine?**				
*1 or 2 meals a day*	247 (21.9%)	146 (20.7%)	101 (23.9%)	0.350 ***
*3–4 meals a day*	761 (67.3%)	480 (67.9%)	281 (66.4%)	
*5 or more meals a day*	122 (10.8%)	81 (11.5%)	41 (9.7%)	
**How many meals do you eat in this period of quarantine?**				
*1 or 2 meals a day*	181 (16.0%)	94 (13.3%)	87 (20.6%)	0.002 ***
*3–4 meals a day*	779 (68.9%)	496 (70.2%)	283 (66.9%)	
*5 or more meals a day*	170 (15.0%)	117 (16.5%)	53 (12.5%)	
**Are you consuming more food than before the quarantine?**				
*No*	441 (39.0%)	239 (33.8%)	202 (47.8%)	<0.001 ***
*Yes*	313 (27.7%)	199 (28.1%)	114 (27.0%)	
*Sometimes*	376 (33.3%)	269 (38.0%)	107 (25.3%)	
**Your consumption of fruit and vegetables during this period is?**				
*Increased*	291 (25.8%)	168 (23.8%)	123 (29.1%)	0.140 ***
*Decreased*	96 (8.5%)	63 (8.9%)	33 (7.8%)	
*Similar*	743 (65.8%)	476 (67.3%)	267 (63.1%)	
**Compared to the period before the quarantine, do you consume more convenience food and fast food or do you cook more?**				
*I kept the normal routine*	415 (36.7%)	258 (36.5%)	157 (37.1%)	0.429 ****
*More food cooked at home*	702 (62.1%)	443 (62.7%)	259 (61.2%)	
*More convenience food and fast food*	13 (1.2%)	6 (0.8%)	7 (1.7%)	
**Compared to the period before the quarantine, do you think you have improved your diet and the quality of the food you eat?**				
*No, I kept the same eating habits*	643 (56.9%)	439 (62.1%)	204 (48.2%)	<0.001 ***
*No, my eating habits got worse*	155 (13.7%)	90 (12.7%)	65 (15.4%)	
*Yes, my diet has improved*	332 (29.4%)	178 (25.2%)	154 (36.4%)	
**Do you have breakfast?**				
*No*	121 (10.7%)	59 (8.3%)	62 (14.7%)	<0.001 ***
*Yes*	1009 (89.3%)	648 (91.7%)	361 (85.3%)	
**Do you eat during the day also if you are not hungry?**				
*Daily*	257 (22.7%)	165 (23.3%)	92 (21.7%)	0.630 ***
*1–2 times a week*	103 (9.1%)	64 (9.1%)	39 (9.2%)	
*3–4 times a week*	111 (9.8%)	75 (10.6%)	36 (8.5%)	
*Occasionally*	489 (43.3%)	295 (41.7%)	194 (45.9%)	
*Never*	170 (15.0%)	108 (15.3%)	62 (14.7%)	
**Do you make aperitive? (alone or in company on video-call)**				
*Daily*	9 (0.8%)	9 (1.3%)	0 (0.0%)	0.185 ****
*1–2 times a week*	59 (5.2%)	36 (5.1%)	23 (5.4%)	
*3–4 times a week*	12 (1.1%)	7 (1.0%)	5 (1.2%)	
*Occasionally*	322 (28.5%)	201 (28.4%)	121 (28.6%)	
*Never*	728 (64.4%)	454 (64.2%)	274 (64.8%)	
**How much water do you drink per day?**				
*<1.5 L*	476 (42.1%)	331 (46.8%)	145 (34.3%)	<0.001 ***
*≥1.5 L*	654 (57.9%)	376 (53.2%)	278 (5.7%)	

*N* (%) or mean and standard deviation or median (IQR) are shown when appropriate; *** Pearson’s chi-square test or **** Fisher’s exact test.

**Table 5 ijerph-18-01502-t005:** Effects of the lockdown on physical activity habits of the university population and stratification by gender.

	Total *N* = 1130	Female *N* = 707	Male *N* = 423	*p*-Value
**Did you participate in the following activities before quarantine?**				
*None*	274 (24.2%)	178 (25.2%)	96 (22.7%)	<0.001 ****
*Individual run/walk*	224 (19.8%)	133 (18.8%)	91 (21.5%)	
*Running/walking group*	16 (1.4%)	12 (1.7%)	4 (0.9%)	
*Gym*	375 (33.2%)	262 (37.1%)	113 (26.7%)	
*Team sports*	64 (5.7%)	21 (3.0%)	43 (10.2%)	
*Other (to specify)*	177 (15.7%)	101 (14.3%)	76 (18.0%)	
**How much time do you spend to do moderate activities during the week?**				
*< 30 min*	373 (33.0%)	209 (29.6%)	164 (38.8%)	0.002 ***
*30–90 min*	379 (33.5%)	251 (35.5%)	128 (30.3%)	
*90–150 min*	196 (17.3%)	131 (18.5%)	65 (15.4%)	
*150–300 min*	119 (10.5%)	84 (11.9%)	35 (8.3%)	
*> 300 min*	63 (5.6%)	32 (4.5%)	31 (7.3%)	
**How much time do you spend to do vigorous activities during the week**				
*< 30 min*	619 (54.8%)	389 (55.0%)	230 (54.4%)	0.108 ***
*30–90 min*	231 (20.4%)	142 (20.1%)	89 (21.0%)	
*90–150 min*	117 (10.4%)	84 (11.9%)	33 (7.8%)	
*150–300 min*	124 (11.0%)	72 (10.2%)	52 (12.3%)	
*> 300 min*	39 (3.5%)	20 (2.8%)	19 (4.5%)	
**What was your exercise frequency before the quarantine?**				
*None*	358 (31.7%)	222 (31.4%)	136 (32.2%)	0.002 ***
*1 or 2 times a week*	304 (26.9%)	191 (27.0%)	113 (26.7%)	
*3 times a week*	328 (29.0%)	226 (32.0%)	102 (24.1%)	
*4 or 5 times a week*	121 (10.7%)	59 (8.3%)	62 (14.7%)	
*6 o 7 times a week*	19 (1.7%)	9 (1.3%)	10 (2.4%)	
**Compared to this period of quarantine your physical activity:**				
*Increased*	347 (30.7%)	246 (34.8%)	101 (23.9%)	<0.001 ***
*Decreased*	486 (43.0%)	299 (42.3%)	187 (44.2%)	
*The same*	297 (26.3%)	162 (22.9%)	135 (31.9%)	
**What is your current exercise frequency during the quarantine?**				
*None physical activity*	326 (28.8%)	181 (25.6%)	145 (34.3%)	0.030 ***
*1 or 2 times a week*	318 (28.1%)	203 (28.7%)	115 (27.2%)	
*3 times a week*	240 (21.2%)	160 (22.6%)	80 (18.9%)	
*4 or 5 times a week*	170 (15.0%)	115 (16.3%)	55 (13.0%)	
*6 or 7 times a week*	76 (6.7%)	48 (6.8%)	28 (6.6%)	
**What goal do you want to achieve through physical activities?**				
*Increasing general physical performances*	332 (32.0%)	198 (30.4%)	134 (34.8%)	<0.001 ***
*Increasing sport performances*	53 (5.1%)	19 (2.9%)	34 (8.8%)	
*Losing weight*	307 (29.6%)	222 (34.0%)	85 (22.1%)	
*Improving the state of health*	345 (33.3%)	213 (32.7%)	132 (34.3%)	
**Are you following exercises tutorial on internet?**				
*No*	608 (53.8%)	313 (44.3%)	295 (69.7%)	<0.001 ***
*Yes*	522 (46.2%)	394 (55.7%)	128 (30.3%)	
**If you answered yes (previous question), indicate which**				
*Physical exercise applications*	161 (30.6%)	129 (32.3%)	32 (25.0%)	0.010 ****
*Facebook*	18 (3.4%)	13 (3.3%)	5 (3.9%)	
*Instagram*	59 (11.2%)	53 (13.3%)	6 (4.7%)	
*YouTube*	243 (46.1%)	174 (43.6%)	69 (53.9%)	
*Other (to specify)*	46 (8.7%)	30 (7.5%)	16 (12.5%)	
**Do you train alone during the quarantine?**				
*No*	298 (31.8%)	199 (33.0%)	99 (29.6%)	0.280 ***
*Yes*	640 (68.2%)	404 (67.0%)	236 (70.4%)	
**Do you believe that the quarantine period has affected the amount of time you spend sitting or lying down?**				
*No*	111 (9.8%)	63 (8.9%)	48 (11.3%)	0.081 ***
*Yes, I spend less time sitting or lying down*	39 (3.5%)	30 (4.2%)	9 (2.1%)	
*Yes, I spend more time sitting or lying down*	980 (86.7%)	614 (86.8%)	366 (86.5%)	

*N* (%) or mean and standard deviation or median (IQR) are shown when appropriate; *** Pearson’s chi-square test or **** Fisher’s exact test.

## Data Availability

In this section, please provide details regarding where data supporting reported results can be found, including links to publicly archived datasets analyzed or generated during the study. Please refer to suggested Data Availability Statements in section “MDPI Research Data Policies” at https://www.mdpi.com/ethics. You might choose to exclude this statement if the study did not report any data.

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
