# Peer review of "Effects of the COVID-19 Pandemic on Job Activity, Dietary Behaviours and Physical Activity Habits of University Population of Naples, Federico II-Italy"

_ijerph, 2021, doi:10.3390/ijerph18041502_

Round 1
Reviewer 1 Report
Congratulations on good work. Interesting paper, with some novel results, in this new Pandemic time. Despite the novelty, the study is observational and simple. In my opinion, there are some issues that should be clarified /improved. Methods section should be improved, more information is needed about the subjects selection and questionnaire implementation. Bias problems could exist.
Some specific comments are provided below:
Abstract:
- Seems too extensive and should be reduced;
- A little confuse about conclusion. This is also because no aim is presented by the authors. Please, clarify
Introduction
- It seems that the introduction is too focused on the measures provided by the Italian government. Perhaps it should focus on studies regarding isolation times, in other contexts, to justify the relevance of this research. It lacks the rationale.
- Please also provide hypothesis for the study in the end of introduction.
Material and methods:
- Some characteristics of the group could be provided here and not in results, for example, age, and age range.
- How was the selection performed?
- Were there any inclusion/exclusion criteria?
- Was there any calculation for the number of subjects needed? (Sample size)?
- How was the questionnaire implemented? How long? Alone, together? ... it is important for the authors to clarify all the procedures so that the study could be replicated. This is essential.
Results
- Tables could be improved, so that could be easily understood. For instance, there are too many horizontal lines. Moreover, is it possible to provide some Graphs to replace some Tables and to better understand information?
- Did the authors compare before quarantine with after quarantine? Please clarify, as I did not see this in the results and I think this is important to design discussion and conclusion
Discussion
- Discussion must be improved. There are many paragraphs with a single sentence and the ideas are not linked between each other. The authors should make an effort to better discuss data with previous results (from other similar situations) and provide causes and consequences of the results.
- Was the mean gains and differences significant?
- Should we trust in the people auto-answers?
- The authors should provide limitations for the study.
Author Response
Dear Editor,
thank you for the Reviewer’s Report about our manuscript entitled “Effects of the COVID-19 pandemic on job activity, dietary behaviours and physical activity habits of University population of Naples, Federico II -Italy”, submitted to International Journal of Environmental Research and Public Health (IJERPH).
We have appreciated the comments received by the Reviewers and have carefully considered them in preparing a new version of the manuscript. A point-by-point response to the comments is attached below.
We believe that the manuscript is now significantly improved thanks to the Reviewer’s inputs.
We hope that the new version of the paper deserve publication on the Journal of Environmental Research and Public Health.
Best regards,
Prof. Dr. Olga Scudiero
Point-by-point response.
Reviewer 1
Comments and Suggestions for Authors
Congratulations on good work. Interesting paper, with some novel results, in this new Pandemic time. Despite the novelty, the study is observational and simple. In my opinion, there are some issues that should be clarified /improved. Methods section should be improved, more information is needed about the subjects selection and questionnaire implementation. Bias problems could exist.
Some specific comments are provided below:
Abstract:
- Seems too extensive and should be reduced;
- A little confuse about conclusion. This is also because no aim is presented by the authors. Please, clarify
Thanks for the suggestions.
In order to satisfy your request, we made changes to the abstract, underlining the purpose of the work.
Changes are underlined in yellow.
Introduction
- It seems that the introduction is too focused on the measures provided by the Italian government. Perhaps it should focus on studies regarding isolation times, in other contexts, to justify the relevance of this research. It lacks the rationale.
- Please also provide hypothesis for the study in the end of introduction.
Thanks for the suggestions.
In order to satisfy your request, we made changes to the introduction. First, from line 75 to line 76 we have reported 3 new references (numbers 11,12,13) in order to compare our study with other. Secondly, the hypothesis/aims are reported from line 87 to line 91. Changes are underlined in yellow.
Material and methods:
- Some characteristics of the group could be provided here and not in results, for example, age, and age range.
- How was the selection performed?
- Were there any inclusion/exclusion criteria?
- Was there any calculation for the number of subjects needed? (Sample size)?
- How was the questionnaire implemented? How long? Alone, together? ... it is important for the authors to clarify all the procedures so that the study could be replicated. This is essential.
Thanks for the suggestions.
In order to satisfy your request, we made changes to the section 2.1. Formulation and administration of the questionnaire.
Changes are underlined in yellow.
Results
- Tables could be improved, so that could be easily understood. For instance, there are too many horizontal lines. Moreover, is it possible to provide some Graphs to replace some Tables and to better understand information?
- Did the authors compare before quarantine with after quarantine? Please clarify, as I did not see this in the results and I think this is important to design discussion and conclusion
Thanks for the suggestions.
In order to satisfy your request, we made changes to the results section.
We reported some data on graphs and in the tables we have compared our parameters before and after the quarantine.
Changes are underlined in yellow
Discussion
- Discussion must be improved. There are many paragraphs with a single sentence and the ideas are not linked between each other. The authors should make an effort to better discuss data with previous results (from other similar situations) and provide causes and consequences of the results.
- Was the mean gains and differences significant?
- Should we trust in the people auto-answers?
- The authors should provide limitations for the study.
Thanks for the suggestions.
In order to satisfy your request, we made changes to discussion section.
Also, we have improved the literature Changes are underlined in yellow
Reviewer 2
Comments and Suggestions for Authors
Interesting study by Brancaccio et al., however the introduction and discussion need a profound improvement.
The abstract is too long and should be shortened.
Information regarding Informed consent need to be provided as well as the Ethics Committee which approved this research.
The Introduction and Discussion need to be better supported with the literature. I suggest to deepen these section, focusing on the eating and exercise habits. Little is provided about this.
Thanks for the suggestions.
In order to satisfy your request, we made changes to abstract, introduction, material and methods, results and discussion sections.
Also, we have improved the literature.
Moreover, the approval of the Ethics Committee was not requested because the study was conducted through an online survey. Consent to participation was provided online and made explicit in the information sent and read before providing consent.
In addition, the questionnaire was distributed through the university portal and administered anonymously. Finally, the link to the questionnaire presents the following legislation regarding the processing of personal data:
CONFIDENTIALITY OF DATA PROVIDED WITH THIS QUESTIONNAIRE
According to Law 675/1996 and the subsequent Legislative Decree 196/2003, all information collected with the questionnaires will be used exclusively for scientific research purposes (Article 12, c. 1, point d). Furthermore, the data collected in the context of this survey are protected by statistical secrecy and therefore cannot be communicated or externalized except in aggregate form, so that no individual reference can be made, and can only be used for purposes statistics (Article 9 of Legislative Decree No. 322 of 6 September 1989). Finally, the data collected will be made anonymous, during the computer processing, pursuant to art. 1, c. 2 point i) of law 675/1996.
Changes are underlined in yellow.
Reviewer 3
Comments and Suggestions for Authors
The manuscript entitled “Effects of the COVID-19 pandemic on job activity, dietary behaviours and physical activity habits of University population of Naples, Federico II -Italy” presents interesting issue, but some areas must be corrected.
Major:
- The major problem with the presented manuscript is associated with the fact that Authors prepared it based on mainly their national Italian references, as a vast majority of references presents only their national perspective, not international one. Authors should prepare manuscript not only to be interesting for Italian readers, but to be interesting for international readers. If Authors prepare their manuscript only for their national readers, they should publish it in some national journal. So, Authors should present international data from various countries, and international perspective not only the Italian one.
- Moreover, over a quarter of references are the self-citations of Authors own publications, being also an ethical issue, as Authors should not refer the publications which are not associated with the aim and scope of their study (e.g. studies conducted in groups of athletes or associated with childhood obesity – not associated with COVID-19 pandemic at all – ref. 16, 21, 22, 24, 25, 26, 27, 28 – those references should be removed and replaced by more adequate ones)
Thanks for the suggestions.
In order to satisfy your request, we have made changes both in the text and in the literature.
Changes are highlighted in yellow.
General:
Authors present some basic and even very trivial information not formulated while using a proper scientific language, that should not be presented in a scientific manuscript (e.g. “The new coronavirus, which has started to spread since December 2019, has brought the world community to its knees.”) – Authors should be aware that they do not prepare the column of the newspaper, but a scientific paper that should be interesting for researchers.
Authors should properly prepare their manuscript based on the instruction for authors
Thanks for the suggestions.
In order to satisfy your request, we have made changes both in the text.
Changes are highlighted in yellow.
Abstract:
Lines 25-28 – should be removed as they do not present any justification of the presented paper, but only a basic information
Authors should properly formulate the aim of their study in this section (e.g. “The aim of the study was…”), instead of only indicating what was done within the study.
Authors should briefly describe applied questionnaire (was it validated?).
Lines 46-47 – should be removed as they do not present any conclusions of the presented paper, but only self-appraisal
Thanks for the suggestions.
In order to satisfy your request, we have made changes in the abstract.
Changes are highlighted in yellow.
Introduction:
Lines 53-64 – basic information about COVID should not be presented here
Lines 65-78 – presentations of the Italian situation during the study should be transferred to Materials and Methods Section
Lines 79-81 – such information should be the main part of the Section, but Authors should present more specific information (observations from other studies and other countries) – there are a lot of such studies, even a simple Pubmed search reveals hundreds of various studies (see examples below)
Authors should properly present the international perspective for job activity, dietary behaviours and physical activity habits – they should describe results observed by other authors during the period of COVID, especially in the populations of students (as described above), e.g.:
- job activity – e.g. https://www.ncbi.nlm.nih.gov/pmc/articles/PMC7482653/, https://www.ncbi.nlm.nih.gov/pmc/articles/PMC7674395/, https://www.ncbi.nlm.nih.gov/pmc/articles/PMC7451187/,
- dietary behaviours - https://www.ncbi.nlm.nih.gov/pmc/articles/PMC7352706/, https://www.ncbi.nlm.nih.gov/pmc/articles/PMC7551462/, https://www.ncbi.nlm.nih.gov/pmc/articles/PMC7766569/, https://www.ncbi.nlm.nih.gov/pmc/articles/PMC7353108/
- physical activity habits - https://www.ncbi.nlm.nih.gov/pmc/articles/PMC7526007/, https://www.ncbi.nlm.nih.gov/pmc/articles/PMC7605130/, https://www.ncbi.nlm.nih.gov/pmc/articles/PMC7384265/, https://pubmed.ncbi.nlm.nih.gov/32557827/
Thanks for the suggestions.
In order to satisfy your request, we have made changes both in the text and in the literature.
The papers he/she suggested have also been added.
Changes are highlighted in yellow.
Materials and Methods:
Authors should clearly state if they received the ethical committee agreement for their study.
Thank you for that observation. The approval of the Ethics Committee was not requested because the study was conducted through an online survey. Consent to participation was provided online and made explicit in the information sent and read before providing consent
As they conducted anonymized study, they did not have to do it, but they must clearly indicate what was their approach.
Thank you for that observation. As defined in the paragraph of the methods , the study was conducted online and anonymously
Authors should clearly describe applied procedure (e.g. inclusion, exclusion criteria, verification of the representativeness of the studied group, distribution of the questionnaire, etc)
Authors should clearly describe applied tool (questionnaire), with all necessary details (e.g. questions that were asked, was it validated, what type of questions were asked, what was the form of the questionnaire, etc.)
Thanks for the suggestions. In order to satisfy your request, we made changes to section 2.1. Formulation and administration of the questionnaire.Changes are underlined in yellow
It seems that Authors treated all the variables as normally distributed, but based on the presented mean and SD values, it may be supposed that data for some variables were characterised by the distribution different than normal.
Authors should (1) verify the normality of distribution, (2) for normally distributed data present mean and SD values, but for the other distributions – present median, min and max values, (3) apply adequate statistical tests, that are based on the distribution.
Normality distribution for quantitative variables was assessed by the Shapiro-Wilk and we have inserted what has been reported.
Results:
Authors should not reproduce in the text information that are already presented in tables.
Tables should be self-explanatory – the distribution of data should be presented
The results section has been implemented by figures to support when described in the tables.
Furthermore, the statistical analysis carried out as well as the distribution have been explained in the materials and methods.
Discussion:
Authors should not reproduce the results in this section.
Authors should not refer tables in this section.
Authors should in their discussion include 3 areas: (1) compare gathered data with the results by other authors, (2) formulate implications of the results of their study and studies by other authors, (3) formulate the future areas which should be studied (see above)
Authors should discuss limitations of their study
Thanks for the suggestions.
In order to satisfy your request, we have made changes in discussion section
Changes are highlighted in yellow.
Conclusions:
Lines 237-239, 244-246 – should be removed as they do not present any conclusions of the presented paper
Thanks for the suggestions.
In order to satisfy your request, we have made changes in conclusions section
Changes are highlighted in yellow.
Author Contributions:
It seems that contribution of some Authors (LC, CFB, CF, CM) was only minor and they did not participate in preparing manuscript. There is a serious risk of a guest authorship procedure which is forbidden. In such case they should be rather presented in Acknowledgements Section and not be indicated as authors of the study.
Thanks for the suggestions.
The authors L.C, C.F.B., C.F., C.M (Cristina Montomoli) are involved in the formulation of questionnaire and at the same time, they contributed to revised the manuscript.
We have reported the functions of each participant in the "contributions of the authors" section as required by the guidelines of the journal during the submission procedure.
In addition, all authors have authorized the submission of the manuscript.

Reviewer 2 Report
Interesting study by Brancaccio et al., however the introduction and discussion need a profound improvement.
The abstract is too long and should be shortened.
Information regarding Informed consent need to be provided as well as the Ethics Committee which approved this research.
The Introduction and Discussion need to be better supported with the literature. I suggest to deepen these section, focusing on the eating and exercise habits. Little is provided about this.
Author Response

(The authors gave the same response as above.)

Reviewer 3 Report
The manuscript entitled “Effects of the COVID-19 pandemic on job activity, dietary behaviours and physical activity habits of University population of Naples, Federico II -Italy” presents interesting issue, but some areas must be corrected.
Major:
- The major problem with the presented manuscript is associated with the fact that Authors prepared it based on mainly their national Italian references, as a vast majority of references presents only their national perspective, not international one. Authors should prepare manuscript not only to be interesting for Italian readers, but to be interesting for international readers. If Authors prepare their manuscript only for their national readers, they should publish it in some national journal. So, Authors should present international data from various countries, and international perspective not only the Italian one.
- Moreover, over a quarter of references are the self-citations of Authors own publications, being also an ethical issue, as Authors should not refer the publications which are not associated with the aim and scope of their study (e.g. studies conducted in groups of athletes or associated with childhood obesity – not associated with COVID-19 pandemic at all – ref. 16, 21, 22, 24, 25, 26, 27, 28 – those references should be removed and replaced by more adequate ones)
General:
Authors present some basic and even very trivial information not formulated while using a proper scientific language, that should not be presented in a scientific manuscript (e.g. “The new coronavirus, which has started to spread since December 2019, has brought the world community to its knees.”) – Authors should be aware that they do not prepare the column of the newspaper, but a scientific paper that should be interesting for researchers.
Authors should properly prepare their manuscript based on the instruction for authors
Abstract:
Lines 25-28 – should be removed as they do not present any justification of the presented paper, but only a basic information
Authors should properly formulate the aim of their study in this section (e.g. “The aim of the study was…”), instead of only indicating what was done within the study.
Authors should briefly describe applied questionnaire (was it validated?).
Lines 46-47 – should be removed as they do not present any conclusions of the presented paper, but only self-appraisal
Introduction:
Lines 53-64 – basic information about COVID should not be presented here
Lines 65-78 – presentations of the Italian situation during the study should be transferred to Materials and Methods Section
Lines 79-81 – such information should be the main part of the Section, but Authors should present more specific information (observations from other studies and other countries) – there are a lot of such studies, even a simple Pubmed search reveals hundreds of various studies (see examples below)
Authors should properly present the international perspective for job activity, dietary behaviours and physical activity habits – they should describe results observed by other authors during the period of COVID, especially in the populations of students (as described above), e.g.:
- job activity – e.g. https://www.ncbi.nlm.nih.gov/pmc/articles/PMC7482653/, https://www.ncbi.nlm.nih.gov/pmc/articles/PMC7674395/, https://www.ncbi.nlm.nih.gov/pmc/articles/PMC7451187/,
- dietary behaviours - https://www.ncbi.nlm.nih.gov/pmc/articles/PMC7352706/, https://www.ncbi.nlm.nih.gov/pmc/articles/PMC7551462/, https://www.ncbi.nlm.nih.gov/pmc/articles/PMC7766569/, https://www.ncbi.nlm.nih.gov/pmc/articles/PMC7353108/
- physical activity habits - https://www.ncbi.nlm.nih.gov/pmc/articles/PMC7526007/, https://www.ncbi.nlm.nih.gov/pmc/articles/PMC7605130/, https://www.ncbi.nlm.nih.gov/pmc/articles/PMC7384265/, https://pubmed.ncbi.nlm.nih.gov/32557827/
Materials and Methods:
Authors should clearly state if they received the ethical committee agreement for their study. As they conducted anonymized study, they did not have to do it, but they must clearly indicate what was their approach.
Authors should clearly describe applied procedure (e.g. inclusion, exclusion criteria, verification of the representativeness of the studied group, distribution of the questionnaire, etc)
Authors should clearly describe applied tool (questionnaire), with all necessary details (e.g. questions that were asked, was it validated, what type of questions were asked, what was the form of the questionnaire, etc.)
It seems that Authors treated all the variables as normally distributed, but based on the presented mean and SD values, it may be supposed that data for some variables were characterised by the distribution different than normal.
Authors should (1) verify the normality of distribution, (2) for normally distributed data present mean and SD values, but for the other distributions – present median, min and max values, (3) apply adequate statistical tests, that are based on the distribution.
Results:
Authors should not reproduce in the text information that are already presented in tables.
Tables should be self-explanatory – the distribution of data should be presented
Discussion:
Authors should not reproduce the results in this section.
Authors should not refer tables in this section.
Authors should in their discussion include 3 areas: (1) compare gathered data with the results by other authors, (2) formulate implications of the results of their study and studies by other authors, (3) formulate the future areas which should be studied (see above)
Authors should discuss limitations of their study
Conclusions:
Lines 237-239, 244-246 – should be removed as they do not present any conclusions of the presented paper
Author Contributions:
It seems that contribution of some Authors (LC, CFB, CF, CM) was only minor and they did not participate in preparing manuscript. There is a serious risk of a guest authorship procedure which is forbidden. In such case they should be rather presented in Acknowledgements Section and not be indicated as authors of the study.
Author Response

(The authors gave the same response as above.)

Round 2
Reviewer 1 Report
I would like to congratulate the authors for the good work, improving the quality of the manuscript.
Author Response
Dear Editor,
thank you for the Reviewer’s Report about our manuscript entitled “Effects of the COVID-19 pandemic on job activity, dietary behaviours and physical activity habits of University population of Naples, Federico II -Italy”, submitted to International Journal of Environmental Research and Public Health.
The manuscript has been carefully revised in response to the reviewers’ comments and suggestions, which have significantly improved it.
A point by point response is attacked below.
We hope that the new version of the paper deserve publication on The International Journal of Environmental Research and Public Health.
Best regards,
Prof. Dr. Olga Scudiero
Point-by-point response.
Reviewer 1
I would like to congratulate the authors for the good work, improving the quality of the manuscript.
Response:
Thanks to the Reviewer 1 for the helpful comments.
Reviewer 2
The authors have addressed all my comments. So, the manuscript can now be published.
Response:
Thanks to the Reviewer 1 for the helpful comments.
Reviewer 3
The manuscript entitled “Effects of the COVID-19 pandemic on job activity, dietary behaviours and physical activity habits of University population of Naples, Federico II -Italy” presents interesting issue, but some areas must be corrected.
Response:
Thanks to the Reviewer 3 for the helpful comments.
General:
Authors should properly prepare their manuscript based on the instruction for authors
Response:
Thanks to the Reviewer 3 for the helpful comments.
In order to satisfy your request, we have modified text, in particular the abstract and references according to the author’s instructions.
(https://www.mdpi.com/journal/ijerph/instructions)
Abstract:
Authors should properly formulate the aim of their study in this section (e.g. “The aim of the study was…”), instead of only indicating what was done within the study.
Response:
Thanks to the Reviewer 3 for the helpful comments.
The aim of the study is reported in abstract underlined in turquoise.
Authors should briefly describe applied questionnaire (was it validated?).
The questionnaire was described in the materials and methods section, it was not validated.
Materials and Methods:
Authors should clearly state if they received the ethical committee agreement for their study. As they conducted anonymized study, they did not have to do it, but they must clearly indicate what was their approach.
Authors should clearly describe applied tool (questionnaire), with all necessary details (e.g. questions that were asked, was it validated, what type of questions were asked, what was the form of the questionnaire, etc.). If questionnaire was not validated previously, it should be indicated here and as a limitation of the study.
Response:
Thanks to the Reviewer 3 for the helpful comments.
In green we have reported the changes required in the materials and methods section.
Results:
Figures are hard to follow – Authors should rather present their data as tables
Response:
As suggested by reviewer 1 in the previous review, we have introduced some graphics in the manuscript to facilitate and improve understanding of the results.
Discussion:
Authors should not reproduce the results in this section.
Authors should not refer tables in this section.
Authors should discuss limitations of their study (associated with validation of the questionnaire)
Response:
Thanks to the Reviewer 3 for the helpful comments.
We modified the discussion by reporting our considerations in the results section
Finally underlined in turquoise we have reported the limitations of our study as required.
All the changes are underlaying in turquoise

Reviewer 2 Report
The authors have addresses all my comments. So, the manuscript can now be published.
Author Response

(The authors gave the same response as above.)

Reviewer 3 Report
The manuscript entitled “Effects of the COVID-19 pandemic on job activity, dietary behaviours and physical activity habits of University population of Naples, Federico II -Italy” presents interesting issue, but some areas must be corrected.
General:
Authors should properly prepare their manuscript based on the instruction for authors
Abstract:
Authors should properly formulate the aim of their study in this section (e.g. “The aim of the study was…”), instead of only indicating what was done within the study.
Authors should briefly describe applied questionnaire (was it validated?).
Materials and Methods:
Authors should clearly state if they received the ethical committee agreement for their study. As they conducted anonymized study, they did not have to do it, but they must clearly indicate what was their approach.
Authors should clearly describe applied tool (questionnaire), with all necessary details (e.g. questions that were asked, was it validated, what type of questions were asked, what was the form of the questionnaire, etc.). If questionnaire was not validated previously, it should be indicated here and as a limitation of the study.
Results:
Figures are hard to follow – Authors should rather present their data as tables
Discussion:
Authors should not reproduce the results in this section.
Authors should not refer tables in this section.
Authors should discuss limitations of their study (associated with validation of the questionnaire)
Author Response

(The authors gave the same response as above.)
